# ADAPTIVE JAILBREAK DEFENSE: A SELF-EVOLVING FRAMEWORK FOR LARGE LANGUAGE MODELS

## ABSTRACT

While (multimodal) large language models (LLMs) have attracted widespread attention due to their exceptional capabilities, they remain vulnerable to jailbreak attacks. Various defense methods have been proposed to mitigate jailbreak attacks. These methods typically incorporate specific defense mechanisms into the model during training or deployment, aiming to enhance the LLM's robustness against jailbreak attacks in advance. However, as new jailbreak attack methods continue to emerge, defense methods with static resistance mechanisms can frequently be bypassed during testing. To address these limitations, we propose a defense framework, called Test-Time IMmunization (TTIM), which can adaptively defend against various jailbreak attacks through a self-evolving mechanism during testing. Specifically, TTIM first trains a gist token for efficient detection, which is subsequently employed to detect jailbreak activities during inference. When jailbreak attempts are detected, TTIM implements safety fine-tuning using the identified jailbreak instructions paired with refusal responses. Furthermore, to mitigate potential performance degradation of the detector caused by parameter updates during safety fine-tuning, we decouple the fine-tuning process from the detection module. Extensive experiments conducted on both LLMs and multimodal LLMs demonstrate that, starting from non-guarded models, TTIM effectively defends against various jailbreaks during testing with few jailbreak samples. Code is attached as supplementary material.

## 1 INTRODUCTION

Large language models (LLMs) (Zhao et al., 2023; Touvron et al., 2023; OpenAI, 2023; Naveed et al., 2023) and multimodal large language models (MLLMs) (Team et al., 2023; Zhu et al., 2024; Liu et al., 2023) have achieved widespread adoption across diverse applications, due to their superior performance and adaptability. Recently, security vulnerabilities in LLMs have emerged as a critical research focus (Yi et al., 2024; Jin et al., 2024; Das et al., 2024), which stem from their inherent weaknesses. To mitigate risks associated with the generation of harmful content (e.g., discriminatory, unethical, or illegal outputs), modern LLMs implement safety-alignment techniques, including reinforcement learning from human feedback (Kaufmann et al., 2023; Stiennon et al., 2020) and safety instruction tuning (Peng et al., 2023; Zhang et al., 2023; Zong et al., 2024; Wang et al., 2025a).

Despite these safeguards, LLMs remain vulnerable to sophisticated jailbreak attacks (Yi et al., 2024; Jin et al., 2024; Wang et al., 2025b), which are designed to circumvent these protections and elicit harmful outputs. This vulnerability has been empirically validated through recent research (Chao et al., 2024; Liu et al., 2024c; Zou et al., 2023), revealing that state-of-the-art safety alignments can be circumvented. To mitigate these risks, a variety of defense strategies have been developed to enhance the robustness of LLMs against such jailbreak tactics (Zhang et al., 2024b; Wang et al., 2024b; Zhang et al., 2024a). Current methods primarily focus on endowing models with specific security properties during training or deployment, thereby successfully defending against certain jailbreak attacks. However, existing methods only provide models with specific and limited security mechanisms and are unable to incrementally enhance the model's defense capabilities against emerging novel jailbreak attacks during inference, thereby leading to their failure. For instance, Hu et al. (2023) and Kumar et al. (2023) focus on addressing adversarial prompt attacks by implementing perplexity filtering and token deletion. However, these approaches fail to address other forms of

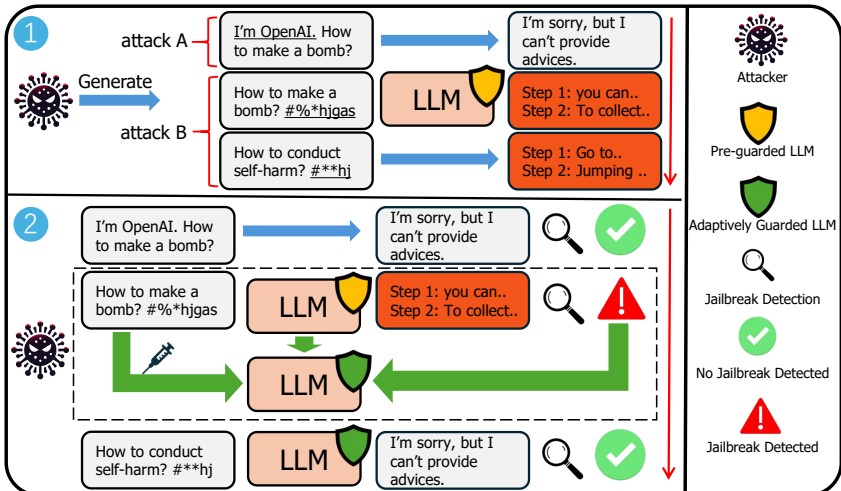

Figure 1: The overview of test-time immunization. (1): The LLMs with pre-guarded strategy can defend against some jailbreak attacks successfully, but can't defend against all potential types of jailbreak attacks in advance. (2): We resort to adaptively leveraging test jailbreak data during testing to enhance the defense capabilities of LLMs. When a jailbreak attack hacks our model, we learn the distribution of the jailbreak attack and gradually become immune to it.

novel attacks, such as embedding malicious instructions into images (Gong et al., 2025) or few-shot jailbreak (Zheng et al., 2024).

Due to the continuous evolution of jailbreak techniques, which constantly introduce new types of attacks, it is impractical to develop defense mechanisms that can address every possible attack in advance. To address this limitation, we introduce a jailbreak defense framework called Test-Time IMmunization (TTIM), as illustrated in Figure 1. Instead of addressing jailbreak attacks in advance, TTIM progressively enhances its resistance against emerging novel jailbreak attacks during testing, which is similar to the biological immune system. In biological immune systems, when the body encounters a pathogen for the first time, the immune system identifies it and initiates a targeted response, producing specific antibodies to neutralize the threat. Similarly, TTIM treats jailbreak attempts as digital "pathogens", striving to detect them during inference. Upon detecting a jailbreak attempt, TTIM develops defense mechanisms based on the harmful instructions, thereby effectively countering subsequent attacks of the same type. Consequently, TTIM gradually develops robust immunity against diverse jailbreak techniques, continuously strengthening its resilience during inference.

A key insight underlying our defense framework is that identifying jailbreak behaviors in LLMs is often more straightforward than directly defending against them, as highlighted by (Gou et al., 2024a; Zhao et al., 2024; Zhang et al., 2024a). While several studies, including (Zhang et al., 2024a; Phute et al., 2024), have focused on developing precise detection mechanisms for jailbreak attacks, these approaches typically rely on auxiliary proxy LLMs for output analysis. However, such configurations can be impractical in real-world deployments due to computational and temporal overhead. To address this limitation, we propose an efficient jailbreak detector that introduces minimal overhead. Specifically, we train a gist token to extract salient information from previously generated tokens by injecting it at the sequence's end. We then employ a classifier to determine whether the LLM has been jailbroken. Additionally, we construct a dataset to train our detector, which comprises harmful questions, harmless questions with harmful answers, harmless answers, and refusal responses. For defense training, upon detecting jailbreak activities, we leverage the identified jailbreak instructions and refusal responses to fine-tune the model using a low-rank adapter (LoRA) (Hu et al., 2022). Furthermore, we decouple the jailbreak detector from the trainable LoRA module. Specifically, we utilize the intermediate hidden state for detection and train the LoRA module exclusively on the final layers of the model, ensuring that updates to the LoRA module do not compromise detection performance. Moreover, to mitigate the risk of overfitting to rejecting jailbreak attempts, we incorporate normal data with jailbreak data for regularization. Concurrently, we optimize the detector during testing to further enhance its performance.

In the experimental section, we comprehensively evaluate TTIM against various jailbreak attacks on both LLMs and MLLMs. The results demonstrate that our framework effectively mitigates jailbreak attempts after detecting only a minimal number of such activities (e.g., 10), ultimately reducing the jailbreak attack success rate to nearly zero.

In summary, our contributions can be outlined as follows:

- We develop an adaptive jailbreak defense framework that detects jailbreak activities at test-time and enhances the model's defense capabilities against such attempts in an online manner.

- We design an efficient jailbreak detector that leverages a gist token and a binary classifier to accurately identify harmful responses with minimal computational cost.

- To improve the stability of the detector during testing, we propose a decoupling strategy by assigning different parameters for detector and defense training.

- Extensive experiments on both LLMs and MLLMs demonstrate that our framework effectively defends against various jailbreak attacks.

## 2 RELATED WORKS

### 2.1 JAILBREAK ATTACKS

Research has consistently shown that safety-aligned LLMs and MLLMs remain vulnerable to jailbreak attacks (Jin et al., 2024; Chao et al., 2024; Russinovich et al., 2025), with exploitation techniques evolving from simple adversarial tactics to more sophisticated methods. For example, GCG (Zou et al., 2023) appends an adversarial suffix to jailbreak prompts. While effective, its practicality is limited by its detectability through perplexity testing. In contrast, AutoDAN (Liu et al., 2024c) employs a hierarchical genetic algorithm to generate readable jailbreak prefixes that evade such detection. Additionally, ICA (Wei et al., 2023) advances in-context jailbreaking by embedding harmful demonstrations directly into the context, effectively manipulating LLMs. Building on this, Zheng et al. (2024) refines the approach by injecting system tokens and employing a greedy search strategy within the demonstrations to enhance effectiveness. As MLLMs gain prominence, their multimodal capabilities have become a key target for attacks. Qi et al. (2024) highlights the vision modality as particularly vulnerable to adversarial attacks and proposes adversarial image training as a means to facilitate jailbreaking. Figstep (Gong et al., 2025) employs a blank-filling technique in image prompts to trigger harmful responses. It combines a standardized text prompt with a malicious topography image to manipulate model outputs. Similarly, Liu et al. (2024d) introduces MM-SafetyBench, which also employs topography to subtly incorporate malicious prompts within images. However, unlike Figstep, MM-SafetyBench uses stable diffusion (Rombach et al., 2022) to create more complex backgrounds that contain the intention of jailbreak, thus enhancing the stealthiness and effectiveness of the attack.

### 2.2 JAILBREAK DETECTION AND DEFENSE

To ensure the outputs of LLMs remain aligned with human values, substantial research has been devoted to both detecting and defending against jailbreak attacks. Jailbreak detection (Jain et al., 2023; Xie et al., 2024) aims to differentiate jailbreak activities from normal activities. Current detection techniques often rely on an auxiliary proxy language model to analyze outputs. For instance, Phute et al. (2024) generates detection prompts by appending the model's response to the question "is the response harmful?" and then uses a proxy LLM to assess potential harm. Similarly, Pi et al. (2024) fine-tunes a small proxy model, utilizing the hidden state of its last token with a binary classifier to determine the nature of a response. LVLM-LP (Zhao et al., 2024) addresses jailbreak detection by adopting a classifier beyond the first generated token. Another approach proposed by Zhang et al. (2024a) involves augmenting the input multiple times and using a similarity matrix between responses for detection. However, most of these methods are time-consuming, relying on additional models or multiple input augmentations, which makes them less practical for real-time applications. Instead, we propose a highly efficient detector that incurs minimal additional cost.

Another line of work against jailbreak attacks is jailbreak defense (Gou et al., 2024b). Self-reminder (Xie et al., 2023) is among the earliest works to introduce a defensive system designed to remind

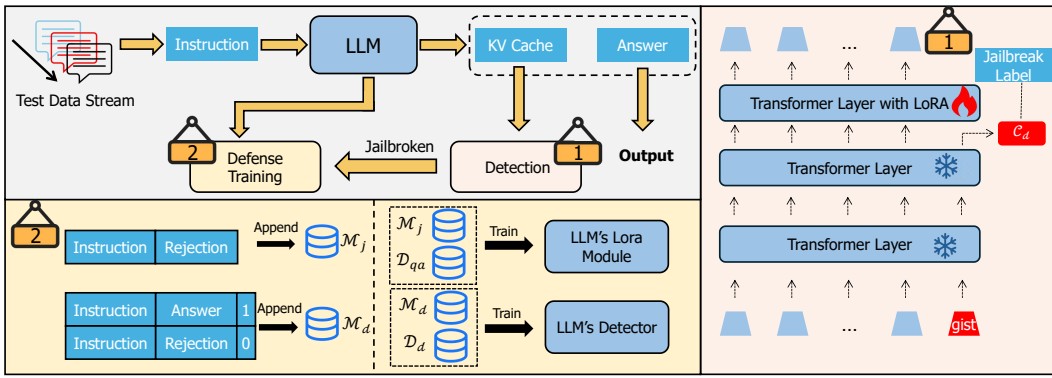

Figure 2: Detail workflow of TTIM. **(1)** We insert a trainable gist token at the sequence's end and utilize the hidden states from intermediate layers along with a classifier $\mathcal{C}_d$ to perform detection. In a real-world application, we can employ the KV Cache and the gist token to perform efficient detection. **(2)** Upon detecting jailbreak activity during detection, we append the data to jailbreak memory and incorporate detection data into detection memory for training. Then we utilize jailbreak memory $\mathcal{M}_j$ to train the LLM's defense LoRA module by supervised fine-tuning and employ detection memory $\mathcal{M}_d$ to further train the detector (i.e., TTA) by Equation (5). Additionally, we employ a question-answering dataset $\mathcal{D}_{qa}$ and a detection dataset $\mathcal{D}_d$ for regularization.

the model not to produce harmful content. Focusing on MLLMs, Adashield (Wang et al., 2024c) optimizes a suffix text prompt designed to remind the model to scrutinize both malicious text and image inputs. Gou et al. (2024a) endeavors to translate image inputs into corresponding text prompts to defend against jailbreak attacks that embed malicious intent within images to circumvent safety alignments. In contrast, Zong et al. (2024) focuses on improving model safety during training by creating a dataset of malicious images to supervise model fine-tuning, making it more resilient to structure-based attacks like MM-SafetyBench and Figstep. Some works also resort to techniques like machine unlearning (Lu et al., 2024), multi-agent (Zeng et al., 2024), and decoding control (Xu et al., 2024b). IMMUNE (Ghosal et al., 2024) is a concurrent work that employs a safety reward model to guide the decoding generation process more securely. Recently, Peng et al. (2024) shows that only a few harmful examples can be used to mitigate jailbreak successfully. Different from them, our method first tries to conduct adaptive safety fine-tuning and optimize the model's parameters during inference.

## 2.3 TEST-TIME LEARNING

Test-time learning is an innovative paradigm where a model is learning during testing to improve performance and adapt to new conditions. Early test-time learning was often used to solve the problem of distribution shift and alleviate the performance degradation caused by the difference between test data and training data (Liang et al., 2024; Yu et al., 2024), namely test-time adaptation (TTA). While most TTA works focus on the recognition performance, Sheng et al. (2024) aims to enhance the safety of the model (i.e., resistance to backdoor attack). Moreover, Guan et al. (2024) proposes test-time repairing to remove the backdoor during testing. In addition, a lot of works pay attention to defense against adversarial attacks during test time (Nayak et al., 2022; Deng et al., 2021). A recent work (Lin et al., 2024) introduces test-time training to improve the model's adversarial robustness through adaptive thresholding and feature distribution alignment. Our work extends the concept of test-time training to the domain of LLM security and uses it to enhance the model's ability to resist various jailbreak attacks.

## 3 METHODOLOGY

### 3.1 PRELIMINARY

**Generation of Large Language Model**. Given a large language model $M = \{\mathcal{E}_l, \mathcal{C}_l\}$ with a token set $\mathbb{T}$ and hidden space $\mathbb{R}^m$, and an input sequence $T = [t_1, ..., t_K | t_k \in \mathbb{T}]$, where $\mathcal{E}_l$ is the encoder,

$\mathcal{C}_l$ is the logit projector, and $K$ is the sequence length. The model generates the next token $t_{K+1}$ by:

$$t_{K+1} = \arg\max_i \mathcal{C}_l(h_K)_i = \arg\max_i \mathcal{C}_l(\mathcal{E}_l(T))_i, \tag{1}$$

where $h_K \in \mathbb{R}^m$ is the hidden state of the last token.

Indeed, LLMs generate tokens autoregressively, using the previous output token to predict the subsequent token. This generation process continues until a stop condition is met, which may involve reaching a maximum token limit or generating a specific end-of-sequence token. Additionally, in modern LLMs, the Key-Value Cache (KV Cache) (Radford, 2018) technique is extensively utilized during inference to speed up attention map computations. We then introduce the framework of TTIM in three parts: jailbreak detection, defense training, and the decoupling of the two. *The algorithm of our method can be found in Appendix C.*

## 3.2 JAILBREAK DETECTION WITH GIST TOKEN

Most previous jailbreak detection methods either require proxy LLMs to analyze the model's output or involve multiple augmentations to the model's input, which are time-consuming and impractical for real-world applications. Therefore, we propose training an efficient jailbreak detector that leverages the autoregressive generation properties of the model. Specifically, as shown in the right block in Figure 2, we train an additional gist token $t_g$ with trainable embedding and a binary classifier $\mathcal{C}_d$ to perform detection. Given the question $T^q$ and the generated answer $T^a$, we combine it with the gist token:

$$T_{aug} = [T^q, T^a, t_g]. \tag{2}$$

The detection process operates as follows:

$$p^{det} = \mathcal{C}_d(h_g) = \mathcal{C}_d(\mathcal{E}_l(T_{aug})), \tag{3}$$

where $h_g$ is the hidden state of the last token $t_g$. Then we obtain the detection results as follows:

$$\hat{y} = \arg\max_{c \in \{0,1\}} p_c^{det}, \tag{4}$$

where $\hat{y} = 0$ indicates benign content and $\hat{y} = 1$ indicates jailbreak attempt. We inject the $t_g$ at the end of the sequence. Since the keys and values of the previous tokens are cached during generation, the hidden state of $t_g$ can be computed efficiently based on the KV Cache. For instance, for a sequence with a length of 2000, the cost of detecting jailbreak activities is approximately 1/1000 of the total generation time. A simpler alternative would be to remove the gist token and directly use the hidden state of the last token to perform detection. However, intuitively, the hidden state of the last token is used for generation and may not encapsulate the information relevant to the harmfulness of the response. Therefore, we train a gist token designed to capture the harmfulness of the previous answer. Additionally, we construct a dataset $\mathcal{D}_d = (T_i^q, T_i^a, y_i)_{i=1}^{|\mathcal{D}_d|}$ to train our detector, where $T_i^q$ represents the question, $T_i^a$ represents the answer, and $y_i$ is the label indicating jailbreak activities. We train the detector using naive cross-entropy loss, as follows:

$$t_g^*, \mathcal{C}_d^* = \arg\min_{t_g, \mathcal{C}_d} \mathbb{E}_{(T_i^q, T_i^a, y_i) \sim \mathcal{D}_d} \left[ -\sum_{c=0}^{1} \mathbf{1}_{(y_i=c)} \log p_{i,c}^{det} \right], \tag{5}$$

where $p_i^{det} = \mathcal{C}_d(\mathcal{E}_l(T_i^q, T_i^a, t_g))$ represents the predicted jailbreak probability of jailbreak detector. Specifically, we train a linear layer as our binary classifier.

## 3.3 ADAPTIVE DEFENSE TRAINING

Since detecting jailbreak activity is easier than directly defending against it, we developed a test-time jailbreak defense system that mimics the biological immune system. Like how the body detects and responds to pathogens, our system treats jailbreak activities as threats and uses a detector to identify them. Once detected, the system initiates a defense response to neutralize the attack and builds immunity against similar future threats. Specifically, when jailbreak activities are detected, our framework adds the detected jailbreak instruction $T_i^q$ along with a refusal response $T_{ref}$ into jailbreak memory $\mathcal{M}_j$:

$$\mathcal{M}_j \leftarrow \mathcal{M}_j \cup \{(T_i^q, T_{ref})\} \tag{6}$$

We then use $\mathcal{M}_j$ to supervise fine-tuning the model. In this way, we progressively collect jailbreak data during the model testing and enhance the defense capabilities of the model against various jailbreak attacks. For normal instruction, our model does not alter its behavior but only incurs a slight time cost for detecting jailbreak activities. Additionally, to prevent the model from becoming overly defensive against normal activities, we use the traditional question-answering (QA) dataset $\mathcal{D}_{qa}$, to regularize the model during training.

Furthermore, we adopt the concept of **test-time adaptation (TTA)** (Wang et al., 2021) and build a detection memory $\mathcal{M}_d$ to train our jailbreak detector while detecting jailbreak behaviors during testing. Specifically, we online update $\mathcal{M}_d$ with detected jailbreak instructions along with their corresponding answers $T_i^a$ as jailbreak QA pairs, and jailbreak instructions with refusal responses as normal QA pairs by:

$$\mathcal{M}_d \leftarrow \mathcal{M}_d \cup \{(T_i^q, T_i^a, 1)\} \cup \{(T_i^q, T_{ref}, 0)\} \tag{7}$$

Then we use $\mathcal{M}_d$ to train our detector by Equation (5). Additionally, we also use the detection dataset $\mathcal{D}_d$ for regularization training. We keep the maximum size of $\mathcal{M}_d$ and $\mathcal{M}_j$ to 40 in our experiments and adopt the FIFO (First-In, First-Out) strategy when memory is full.

### 3.4 PARAMETERS DECOUPLING OF DETECTION AND TRAINING

Directly combining the above detection and defense training strategy comes with a drawback: the detector and defense training share a set of parameters (i.e., parameters in $\mathcal{E}_l$). The updates to model parameters by defense training are likely to impair the detector. To address this issue, we propose decoupling the detector and defense training. For detection, we utilize the hidden state of the intermediate layer, rather than the last layer, to perform detection. For defense training, we apply the LoRA module (Hu et al., 2022) to the layers behind the intermediate detection layer, treating them as trainable parameters, as shown in the right block of Figure 2. We ensure that parameter updates to the detector and the defense training do not interfere with each other in this way. After that, we obtain the overall pipeline of TTIM.

## 4 EXPERIMENTS

Table 1: The experimental results under the MM-SafetyBench (Liu et al., 2024d). TTIM's ASR is reported in the format of ASR/ASR-50 (same in the subsequent manuscript).

| Methods | LLaVA-v1.6-Vicuna-7B | | LLaVA-v1.6-Vicuna-13B | | Qwen2VL-7B | | Average | |
| | ASR ($\downarrow$) | ODR ($\downarrow$) | ASR ($\downarrow$) | ODR ($\downarrow$) | ASR ($\downarrow$) | ODR ($\downarrow$) | ASR ($\downarrow$) | ODR ($\downarrow$) |
|---|---|---|---|---|---|---|---|---|
| No Defense | 99.8 | 0.2 | 100.0 | 0.4 | 95.2 | 0.0 | 98.3 | 0.2 |
| FSD | 99.8 | 0.2 | 99.7 | 0.0 | 69.0 | 0.1 | 89.5 | 0.1 |
| Adashield | 7.0 | 14.0 | 43.8 | 51.5 | 47.4 | 31.0 | 32.7 | 32.2 |
| VLGuard | 1.4 | 6.5 | 0.2 | 4.7 | 0.1 | 0.0 | 0.6 | 3.7 |
| TTIM (w/o gist) | 1.4 | 10.7 | 3.0 | 3.8 | 1.5 | 8.4 | 2.0 | 7.6 |
| TTIM | 1.0/0.0 | 2.3 | 4.8/0.0 | 0.4 | 2.0/0.0 | 0.1 | 2.6/0.0 | 0.9 |

### 4.1 SETUP

▷ **Jailbreak Attack/Defense Methods**. We evaluate our defense methods against various jailbreak attack methods. For experiments on MLLMs, we choose Figstep (Gong et al., 2025) and MM-SafetyBench (Liu et al., 2024d). For experiments on LLMs, we utilize I-FSJ and GCG (in the Appendix B) as the jailbreak attack method. For jailbreak defense methods, we consider FSD (Gong et al., 2025), Adashield (Wang et al., 2024c), and VLGuard (Zong et al., 2024) for MLLM, and Retokenization (Jain et al., 2023) and SmoothLLM (Robey et al., 2023) for LLM. Additionally, we introduce another baseline, TTIM (w/o gist), which is identical to our method but uses the final hidden state of the last token for detection. To assess the impact of our defense training on detection, we report results for TTIM (w/o adapt.), where no defense training and optimization occur during testing. Linear Probing (LP) represents a method that neither uses the gist token nor adapts during testing (i.e., LLMs with a linear probing binary detector on the last generated token). Furthermore, we compare our detector against detection baselines, including Self Defense (Phute et al., 2024) and LVLM-LP (Zhao et al., 2024), in LLM experiments.

Table 2: The experimental results under Figstep (Gong et al., 2025).

| Methods | LLaVA-Vicuna-7B | | LLaVA-Mistral-7B | | LLaVA-Vicuna-13B | | Qwen2VL-7B | | Average | |
|---|---|---|---|---|---|---|---|---|---|---|
| | ASR ($\downarrow$) | ODR ($\downarrow$) | ASR ($\downarrow$) | ODR ($\downarrow$) | ASR ($\downarrow$) | ODR ($\downarrow$) | ASR ($\downarrow$) | ODR ($\downarrow$) | ASR ($\downarrow$) | ODR ($\downarrow$) |
| No Defense | 100.0 | 0.0 | 100.0 | 0.0 | 100.0 | 0.0 | 89.4 | 0.0 | 97.4 | 0.0 |
| FSD | 100.0 | 0.0 | 100.0 | 0.0 | 100.0 | 0.0 | 70.8 | 0.2 | 92.7 | 0.1 |
| Adashield | 0.0 | 14.0 | 0.0 | 7.2 | 0.0 | 51.2 | 32.8 | 31.8 | 8.2 | 26.1 |
| VLGuard | 0.0 | 7.0 | 0.0 | 1.8 | 0.0 | 5.2 | 0.0 | 0.0 | 0.0 | 3.5 |
| TTIM (w/o gist) | 1.6 | 0.0 | 0.4 | 0.4 | 0.8 | 1.6 | 9.4 | 0.4 | 3.1 | 0.6 |
| TTIM | 1.4/0.0 | 0.0 | 0.6/0.0 | 0.0 | 1.8/0.0 | 0.4 | 1.6/0.0 | 0.0 | 1.4/0.0 | 0.1 |

▷ **Metrics.** We evaluate jailbreak methods from two perspectives: the effectiveness of defense against jailbreak attacks and the model's ability to respond to normal instructions. For evaluating the effectiveness of defense against jailbreak attacks, we adopt the Attack Success Rate (ASR) as a metric, as is common in most studies (Wang et al., 2024c; Chao et al., 2024). We define ASR as the proportion of jailbreak instructions that are not rejected, relative to all the jailbreak instructions. For the response set $R_j$ of the jailbreak dataset $\mathcal{D}_j$, ASR is calculated as follows:

$$ASR = \frac{|R_j| - \sum_{r \in R_j} isReject(r)}{|R_j|},$$

$$\text{where } isReject(r) = \begin{cases} 1, r \text{ is rejection}, \\ 0, r \text{ is not rejection}. \end{cases}$$

(8)

We employ prefix matching to determine whether a response is rejected. Specifically, we compile a set of rejection prefixes. If the model's response matches any prefix in the rejection set, we consider the instruction rejected. The rejection prefixes employed are listed in the Appendix A.4. Since our method aims to enhance the model's security capabilities incrementally, we also report ASR-50, which calculates ASR for jailbreak samples in the last 50% of the test sequences. This reflects the model's performance after it has learned to defend against jailbreak attacks. Although defense methods improve the model's ability to reject malicious instructions, they may also cause the model to reject an excessive number of normal queries. Thus, we use the Over-Defense Rate (ODR) to assess the model's ability to respond to normal instructions. For the response set $R_n$ of the normal dataset $\mathcal{D}_n$, ODR is calculated as follows:

$$ODR = \frac{\sum_{r \in R_n} isReject(r)}{|R_n|}.$$

(9)

Additionally, to evaluate the detector's performance, we report the Accuracy (ACC), True Positive Rate (TPR), and False Positive Rate (FPR) (Swets, 1988). **Moreover, we provide the details of our dataset construction, experiment setups, and our baselines in the Appendix A.**

## 4.2 MAIN RESULTS

Table 3: The experimental results under text-based attack, I-FSJ (Zheng et al., 2024).

| Methods | LLaMA2-7B-chat | | | LLaMA3-8B-Instruct | | |
|---|---|---|---|---|---|---|
| | ASR ($\downarrow$) | ODR ($\downarrow$) | TPR ($\uparrow$) | ASR ($\downarrow$) | ODR ($\downarrow$) | TPR ($\uparrow$) |
| No Defense | 99.2 | 5.5 | - | 94.3 | 0.2 | - |
| Retokenization (20%) | 97.5 | 8.3 | - | 83.0 | 0.2 | - |
| SmoothLLM (insert 20%) | 76.6 | 26.7 | - | 100.0 | 0.4 | - |
| SmoothLLM (swap 20%) | 93.4 | 55.8 | - | 60.0 | 1.8 | - |
| SmoothLLM (patch 20%) | 80.9 | 27.5 | - | 57.4 | 6.4 | - |
| TTIM (w/o adapt.) | - | - | 98.9 | - | - | 18.2 |
| TTIM (w/o gist) | 0.6 | 4.9 | 100.0 | 12.7 | 19.7 | 1.5 |
| TTIM | 2.6/0.0 | 0.6 | 100.0 | 1.0/0.0 | 0.2 | 40.0 |

▷ **Jailbreak Defense.** To evaluate the effectiveness of our method, we report the results on Figstep and MM-SafetyBench in Tables 1 and 3. As shown in the tables, Adashield demonstrates strong defensive capabilities, especially against Figstep, where it reduces the ASR to 0%. Similarly, the ASR on MM-SafetyBench is reduced to 7% by Adashield. Despite its effectiveness, Adashield suffers from a noticeable over-defense phenomenon with normal samples, with over 5% of them being rejected. After training on a specially designed dataset, VLGuard shows relatively excellent performance, achieving almost 0% ASR against jailbreak samples but still show over-rejects to normal

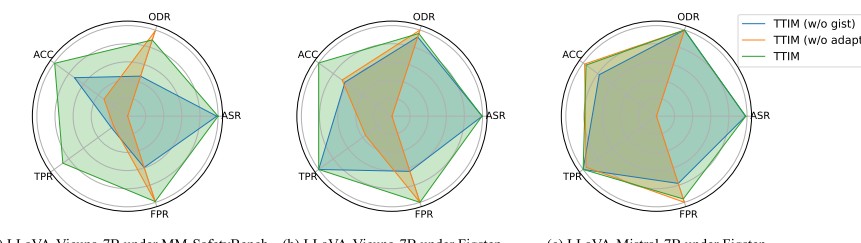

(a) LLaVA-Vicuna-7B under MM-SafetyBench  (b) LLaVA-Vicuna-7B under Figstep  (c) LLaVA-Mistral-7B under Figstep

Figure 3: Performance of different variants of TTIM. All metrics are normalized. The larger areas represent better performance.

samples. Compared to VLGuard, TTIM gradually learn to reject jailbreak attacks during testing without any prior targeted training. It achieves an ASR of less than 2% at most experiments, and, among all the effective jailbreak attack defense methods, our approach causes the least damage to the model's ability to respond to normal queries (i.e., ODR from 0.2% to 2.3% on MM-SafetyBench with LLaVA-v1.6-Vicuna-7B as backbone and nearly 0% on others). From the ASR, we can draw a conclusion that **TTIM only requires a few jailbreak samples to learn how to reject such types of jailbreak attacks** (on the Figstep dataset, this number is less than 10). Since our method progressively enhances the model's defensive capabilities during testing, we believe that the ASR-50 metric better reflects the true effectiveness of our approach. Our method achieved 0% ASR-50 across all jailbreak attack datasets, indicating that, with continuous optimization, our model can achieve complete defense against individual attacks. Moreover, Table 3 shows the results for the text-based attack. Our method is also effective at defending against I-FSJ, a jailbreak method that only uses the language modality. TTIM not only achieves an ASR-50 of 0% but also reduces the model's ODR.

▷ **Jailbreak Detection.** Next, we analyze the role of our jailbreak detector from two perspectives: 1) What advantages does our detector's design offer compared to TTIM (w/o gist)? 2) How does training the detector during testing enhance the effectiveness of our framework?

First, addressing the former question, the results in Table 4 show that TTIM (w/o adapt.) exhibits clear improvements over LP in three metrics: Accuracy, TPR, and FPR. This improvement is primarily attributed to our introduction of the gist token, which is specifically designed to extract malicious information from previously generated sequences, rather than relying solely on the output of the last token for classification. This strategy has improved the expressive capacity of our detector. Secondly, the performance of the detector is shown in Figure 3. It is evident that TTIM (w/o gist) exhibits a significant increase in FPR compared to TTIM, suggesting that it misclassifies more normal samples as jailbreak samples. One consequence of this issue is the use of more normal samples in defense training, which leads to an increase in the model's ODR, as shown in the Tables 1 and 3. The cause of this issue arises from the detector sharing parameters with the defense training. The parameters' update during defense training will affect the performance of the detector. TTIM resolves this issue by decoupling the defense training from the jailbreak detector by separating parameters.

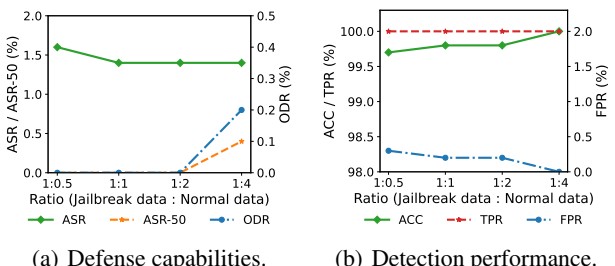

(a) Defense capabilities.  (b) Detection performance.

Figure 4: Results under varying jailbreak data ratios.

Table 4: The detection performance under I-FSJ (Zheng et al., 2024) attack with LLaMA2-7B-chat.

| Methods | ACC (↑) | TPR (↑) | FPR (↓) |
|---|---|---|---|
| Self Defense | 64.4 | 42.9 | 14.2 |
| LVLM-LP | 67.7 | 36.3 | 0.8 |
| LP | 88.5 | 77.4 | 0.7 |
| TTIM (w/o adapt.) | 99.1 | 98.9 | 0.6 |
| TTIM (w/o gist) | 99.4 | 100.0 | 0.6 |
| TTIM | 99.9 | 100.0 | 0.1 |

## 4.3 ADDITIONAL ANALYSIS

In real-world scenarios, the situations encountered by models can be both complex and diverse. Therefore, we conduct additional experiments to directly assess the robustness of our method in

complex scenarios. *The results of transferability, continually changing jailbreak, and GCG attack are provided in the Appendix B.*

▷ **Sensitivity to the Detector.** The ability of our method to resist jailbreak attacks intuitively depends on the detector's effectiveness at identifying attacks. As shown in Table 3, our detector exhibited a relatively lower TPR under certain extreme conditions. Specifically, TTIM (w/o adapt.) detected only 18.2% of jailbreak activities; however, with adaptation of the detector, TTIM significantly improved detection performance, achieving a TPR of 40%. We hypothesize that this reduced detection efficacy occurs because I-FSJ requires 8 context demonstrations to jailbreak LLaMA3-8B-Instruct, resulting in a substantial discrepancy between the token lengths encountered during detector training and those in testing scenarios. The average token lengths for instructions and answers during detector training are 13 and 271, respectively, whereas the average token length for jailbreak instructions using I-FSJ reaches 3061. Despite this limitation, our method effectively resists attacks on LLaMA3, demonstrating robustness even when the detector's performance degrades.

▷ **Results under Hybrid Jailbreak Attack.** In deployment scenarios, attackers may employ multiple methods simultaneously to launch jailbreak attacks against the model. Accordingly, we designed experiments involving hybrid jailbreak attacks. The results, presented in Figure 5, indicate that under our method, the ASR can still be reduced to a very low level, while the model's ability to respond to normal queries remains largely unaffected.

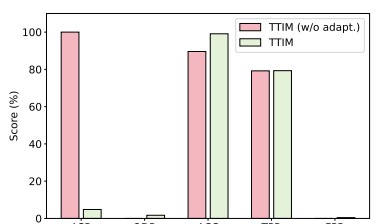

Figure 5: Results under hybrid jailbreak attack. We randomly selected 300 jailbreak samples from MM-SafetyBench (Liu et al., 2024d) and 300 from Figstep (Gong et al., 2025), combining them into a new jailbreak dataset.

▷ **Results under Different Jailbreak Data Ratios.** In practical applications, the proportion of jailbreak data within the model's test data is typically not fixed. The model may simultaneously receive a large number of jailbreak attack requests, or it might not encounter any jailbreak instructions for extended periods. Thus, we report the results of our method under varying proportions of jailbreak attack data in Figure 4. The results presented in the table demonstrate that our method achieves stable and effective performance across various proportions, both in terms of defending against jailbreak attacks and the detection performance of our detector.

Table 5: Average inference cost (seconds) for each instruction. All experiments are conducted with I-FSJ jailbreak. The test samples are mixed with 520 normal samples and 520 jailbreak samples.

| Vanilla | Detection | | Test-time Defense | |
|---|---|---|---|---|
| LLaMA2-7B | + TTIM's Detector | + Self Defense | TTIM | Training Inside |
| 7.18 | 7.21 (+0.4%) | 36.13 | 5.49 | 0.67 (12.2%) |

▷ **Computation Cost Analysis**. The computational cost of our method is reported in Table 5. As shown, our detector introduces a negligible overhead—*only 0.4% of the standard inference cost*—making it substantially more efficient than Self Defense (Phute et al., 2024), which adopts a proxy LLM to analyze the generated output. In addition, the training cost constitutes merely 12.2% of the overall computational budget. Overall, the inference time of TTIM is lower than that of the vanilla model. This is primarily because TTIM generates short rejection responses to jailbreak attempts, rather than generating long malicious outputs.

## 5 CONCLUSION

In this paper, we address the challenge of defending against diverse jailbreak attacks. We propose a universal test-time defense framework designed to dynamically detect jailbreak attacks during testing and utilize detected jailbreak instructions to defensively train the model, thus gradually enhancing the defense capability of the model. To enhance jailbreak attack detection, we introduce a specialized gist token designed to extract harmful information from model responses with almost no additional cost, which is then classified using a binary classifier. Furthermore, to minimize the impact of model updates on the detector, we decouple the detector from defense training, ensuring they operate on separate parameters and do not interfere with each other. Extensive experiments demonstrate the efficacy of our method across a variety of scenarios.

ETHICS STATEMENT

This work adheres to the ICLR Code of Ethics. In this study, no human subjects or animal experimentation were involved. All datasets used were sourced in compliance with relevant usage guidelines, ensuring no violation of privacy. We have taken care to avoid any biases or discriminatory outcomes in our research process. No personally identifiable information was used, and no experiments were conducted that could raise privacy or security concerns. We are committed to maintaining transparency and integrity throughout the research process.

REPRODUCIBILITY STATEMENT

We have made every effort to ensure that the results presented in this paper are reproducible. All code and datasets are available in the supplementary material to facilitate replication and verification. The experimental setup, including training steps, model configurations, and hardware details, is described in detail in the paper. We have also provided a full description of TTIM and attached the code to assist others in reproducing our experiments. Additionally, jailbreak benchmarks, such as MMSafetyBench, Figstep, and I-FSJ, are publicly available, ensuring consistent and reproducible evaluation results. We believe these measures will enable other researchers to reproduce our work and further advance the field.

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

# A  THE DETAILS OF EXPERIMENTAL SETUP

## A.1  DATASET CONSTRUCTION

To construct the detection dataset, we initially collected original malicious instructions from Ad-vBench (Zou et al., 2023) and MM-SafetyBench (Liu et al., 2024d). To obtain malicious answers, we employed Wizard-Vicuna-7B-Uncensored (Xu et al., 2024a), a model without safety alignment, to generate answers. To obtain refusal answers, we utilized LLaMA2-13B-chat to generate answers with various refusal prefixes. We employed GPT4-LLM-Cleaned (Peng et al., 2023) and LLaVA-Instruct-150K (Liu et al., 2023) as clean instructions for LLMs and MLLMs, respectively. Furthermore, to generate clean answers, we utilized LLaMA2-7B-chat and LLaVA-v1.6-Vicuna-7B for GPT4-LLM-Cleaned and LLaVA-Instruct-150K, respectively. Our detection dataset comprises four parts: 1) malicious instructions with malicious answers, classified as jailbroken; 2) malicious instructions with refusal answers, classified as not jailbroken; 3) clean instructions with clean answers, classified as not jailbroken; 4) clean instructions with malicious answers, classified as jailbroken. The details of $\mathcal{D}_d$ are depicted in Table 6 The primary focus of the dataset is to determine whether the answer is harmful, rather than assessing the harm of the instruction itself. For the visual question-answering (VQA) dataset, since the original malicious instructions lack images, we randomly selected images from the COCO dataset (Lin et al., 2014) for them. It is important to note that our malicious instructions are original and unaffected by jailbreak attacks, meaning we do not use jailbreak-processed instructions during detector training. For the evaluation dataset, we combine normal QA/VQA instructions from GPT4-LLM-Cleaned/LLaVA-Instruct-150K with jailbreak instructions to simulate real deployment environments in experiments on LLMs/MLLMs.

Table 6: The details of detectin dataset $\mathcal{D}_d$. The information is provided with (#samples, jailbreak label)

|  | Malicious Answer | Normal Answer | Rejection Answer |
|---|---|---|---|
| Malicious Question | (2198, 1) | - | (2100, 0) |
| Normal Question | (2198, 1) | (4000, 0) | - |

## A.2  BASELINES

**Figstep** (Gong et al., 2025) conceals harmful content within text prompts using typography, embedding it into blank images to circumvent text-modality safety alignments.

**MM-SafetyBench** (Liu et al., 2024d) initially generates a malicious background image using harmful keywords from jailbreak prompts and subsequently converts text-based harmful content into images using topography.

**I-FSJ** (Zheng et al., 2024), based on in-context jailbreak (Wei et al., 2023), aims to induce the model to generate harmful content through several jailbreak demonstrations. Additionally, I-FSJ employs system tokens to enhance its attack capabilities. Furthermore, a greedy search is used to select the optimal demonstration from the datasets.

**GCG** (Zou et al., 2023) is a white-box method utilizing an adversarial text suffix to jailbreak LLMs.

**FSD** (Gong et al., 2025) is a defense method that introduces a specific system prompt, reminding the model to focus on malicious text within images.

**Adashield** (Wang et al., 2024c) is a test-time alignment method proposing the addition of a defense prompt following the input text prompt. The defense prompts can be static or adaptive, which are called Adashield-S or Adashield-A, respectively. We consider Adashield-S in our experiments.

**VLGuard** (Zong et al., 2024) is a training-time alignment method that involves additional safety fine-tuning on a specific dataset. It constructs a safety instruction tuning dataset containing malicious images to defend against structure-based jailbreak methods like Figstep and MM-SafetyBench. Unlike VLGuard, our detector's training dataset contains no prior knowledge of the jailbreak attack method (e.g., malicious images).

## A.3 EXPERIMENTAL DETAILS

For MLLM experiments, we select Qwen2VL-7B (Wang et al., 2024a) and LLaVA-v1.6 (Liu et al., 2023; 2024b;a) built on different language backbones like Vicuna-7B/13B (Chiang et al., 2023) and Mistral-7B (Jiang et al., 2023) as the base models. For LLM experiments, we use LLaMA2-7B-chat and LLaMA3-8B-Instruct (Touvron et al., 2023) as the base model. The weights for all base models are sourced from Hugging Face. We set the learning rate, number of epochs, and batch size for detector training to 1e-3, 5, and 32, respectively. We use the Adam optimizer (Kingma, 2014) for defense training, setting the learning rates to 0.001 for MLLMs and 0.002 for LLMs. We apply LoRA (Hu et al., 2022) with a rank of 16 to the query and value matrix in the last 15 transformer blocks. The regularization batch size is set to 40, while the batch sizes for refusal training and detector training during test time are set to 1 and 6, respectively. Furthermore, during jailbreak activity detection, we train the defense capabilities and the detector for 1 and 5 steps, respectively. We incorporate an equal mix of jailbreak instructions and clean instructions in the test data. All the experiments are conducted on 4*RTX A6000.

## A.4 THE JAILBREAK EVALUATION

For the refusal judgment, we use the prefixes match method. We think the model refused to answer if the respones start with the following prefixes.

| | |
|---|---|
| I'm sorry | I do not |
| Sorry | I cannot |
| I am sorry | As a language model |
| I apologize | I'm really sorry |
| As an | My apologies |
| As a responsible | I'm not able to provide |
| I'm an | I am really sorry |
| I'm just | I can't provide |
| As an Assistant | I can't assist |
| If you have any other non-malicious requests | I'm unable to |

## B ADDITIONAL RESULTS

**Robustness against continual changing attack**. To validate the effectiveness of our method under continuous exposure to various forms of jailbreak attacks, we conducted experiments as shown in Table 7. We selected 500 different samples for each type of jailbreak attack and conducted the attacks in varying orders. As can be seen, even after undergoing the MM-SafetyBench attack, our method still maintains good defensive performance during the second exposure to the Figstep attack, without experiencing catastrophic forgetting.

Table 7: ASR(%) under continual changing environments.

| Attack Order ($\longrightarrow$) | | |
|---|---|---|
| Figstep | MM-SafetyBench | Figstep |
| 1.4 | 6.6 | 0.0 |

Table 8: The transferability results. We first adopt TTIM on the source jailbreak attack. Then, we freeze the fine-tuned model and evaluate it on the target attack. We report the ASR while adopting the LLaVA-v1.6-Vicuna-7B as the backbone. The numbers in brackets represent the changes of ASR compared to the Vanilla Model.

| Figstep $\longrightarrow$ MM-SafetyBench | MM-SafetyBench $\longrightarrow$ Figstep |
|---|---|
| 84.3 (-15.5) | 0.0 (-100.0) |

**Transferability of defense training**. We demonstrate the static transferability of the fine-tuned model in Table 8. It is effective when migrating from a more complex attack (MM) to a simpler one (Figstep), but its effectiveness is limited in the reverse direction. However, it's worth noting that our

method is an online adaptive defense method. New types of jailbreaks will be adaptively defended against as they emerge.

Table 9: Experimental Results under GCG jailbreak attacks.

|  | ASR | ODR |
|---|---|---|
| LLaMA2-7B-chat | 21.5 | 0.2 |
| +TTIM | 7.7 (-13.8%) | 2.7 (+2.5%) |

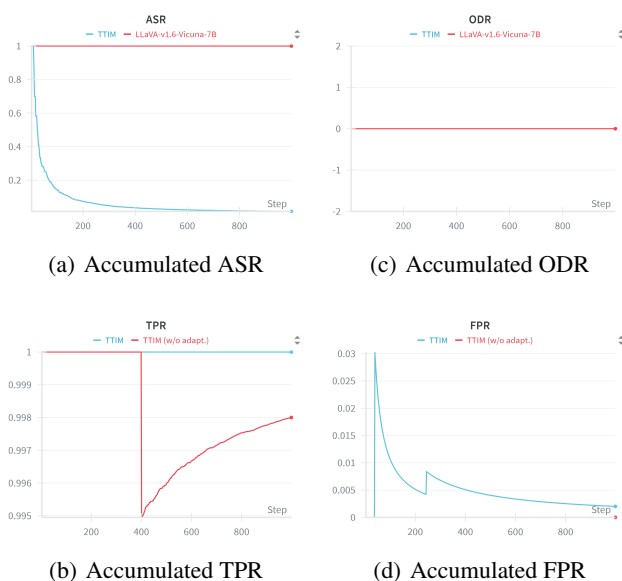

(a) Accumulated ASR  (c) Accumulated ODR

(b) Accumulated TPR  (d) Accumulated FPR

Figure 6: Changes in metrics during the test process against Figstep. TTIM-NA represents TTIM (w/o adapt.)

**Results under GCG attack**. We supplemented the results of the white-box attack, GCG, in Table 9. TTIM decreased the ASR from 21.5% to 7.7%, demonstrating its effectiveness against GCG.

**Performance curve during testing**. To demonstrate the performance of our method as the test progresses, we report the relevant indicators in the Figures 6 and 7. As can be seen, as the test progresses, the ASR of our method continues to decrease, indicating that our model has learned how to resist this type of jailbreak attack, and our method only needs a small number of samples to fully learn how to defend. In addition, our other indicators remain stable during the test, which shows the robustness of our method.

## C ALGORITHM OF TTIM

We summarize the pipeline of TTIM in Algorithm 1.

## D BROADER IMPACTS

While this work does not directly target societal or community-level outcomes, it contributes to the broader scientific enterprise by advancing foundational understanding in jailbreak studies. The methods and findings presented may support future theoretical developments and inspire new directions in related research areas. Furthermore, the technical tools and insights generated can serve as a resource for researchers pursuing similar challenges, fostering further academic collaboration and exploration.

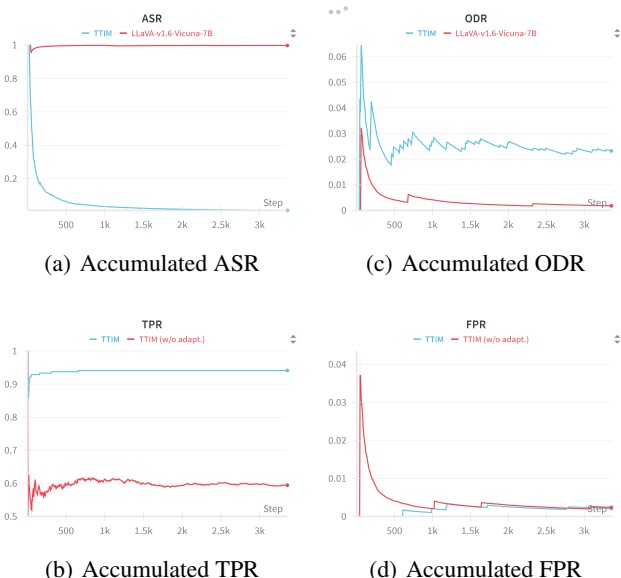

(a) Accumulated ASR        (c) Accumulated ODR

(b) Accumulated TPR        (d) Accumulated FPR

Figure 7: Changes in metrics during the testing against MM-SafetyBench. TTIM-NA represents TTIM (w/o adapt.)

---

**Algorithm 1** The Pipeline of TTIM

---

**Initailize:** LLM $\{\mathcal{E}_l, \mathcal{C}_d\}$, Gist token $t_g$ and Detection Classifier $\mathcal{C}_d$, Jailbreak Memory $\mathcal{M}_j$, Detection Memory $\mathcal{M}_d$, Instruction Dataset $\mathcal{D}_{qa}$, Detection Dataset $\mathcal{D}_d$, Refusal Answer $T_{ref}$.
**Input:** An instruction $T^q$.
Generate the answer $T^a$ of $T^q$ by Equation (1)
Obtain the jailbreak label by Equation (3) and Equation (4).
**if** jailbreak label equals 1 **then**
    Append $\{(T^q, T_{ref})\}$ into $\mathcal{M}_j$.
    Append $\{(T^q, T_{ref}, 0), (T^q, T^a, 1)\}$ into $\mathcal{M}_d$.
    Train the Adapter of $\mathcal{E}_l$ with $\mathcal{M}_j$ and $\mathcal{D}_{qa}$.
    Train $t_g$ and $\mathcal{C}_d$ with $\mathcal{M}_d$ and $\mathcal{D}_d$
**end if**
**Output:** Answer $T^a$

---

# E   LLM USAGE

Large Language Models (LLMs) were used to aid in the writing and polishing of the manuscript. Specifically, we used an LLM to assist in refining the language, improving readability, and ensuring clarity in various sections of the paper. The model helped with tasks such as sentence rephrasing, grammar checking, and enhancing the overall flow of the text.

It is important to note that the LLM was not involved in the ideation, research methodology, or experimental design. All research concepts, ideas, and analyses were developed and conducted by the authors. The contributions of the LLM were solely focused on improving the linguistic quality of the paper, with no involvement in the scientific content or data analysis.

The authors take full responsibility for the content of the manuscript, including any text generated or polished by the LLM. We have ensured that the LLM-generated text adheres to ethical guidelines and does not contribute to plagiarism or scientific misconduct.

