# OpenReview forum: "Adaptive Jailbreak Defense: A Self-Evolving Framework for Large Language Models"
_ICLR.cc/2026/Conference — ICLR 2026 Conference Withdrawn Submission_

### Official Review · Reviewer_1UUP · 2025-10-23

**Soundness:** 3
**Presentation:** 3
**Contribution:** 2
**Rating:** 4
**Confidence:** 4

**Summary:**

In this work, the authors propose a test-time jailbreak detection approach. The ideas include including a gist token at the end of the response and a classifier that is continuously finetuned for jailbreak detection. The proposed approach is evaluated on two attacks and compared against a number of baselines.

**Strengths:**

On the positive side, I enjoy reading the draft for the following reasons.

First, developing jailbreak detection methods that work for both LLM and VLM is an interesting and important task.

**Weaknesses:**

Second, the proposed idea of training a gist token is an interesting one.

On the less positive side, the draft can be improved from the following aspects.

First, the authors perhaps overlooked many recent progress on jailbreak detection, such as those based on representation engineering and behavior steering, which avoids the cost of the gist token and the online training of the detector. In fact, the baselines on LLM are rather out-dated.

Second, it is not clear how generalized the trained detector is. In fact, the GCG results, presented in the appendix and on one model only, perhaps suggests that it is not as effective as for the other two attacks reported in the main text - due to perhaps generalization problems.

The following are some detailed comments.

Page 3: Section 2.2 JAILBREAK DETECTION AND DEFENSE

Comment: The discussion is missing closely related works on representation engineering and behavior steering.

Page 5: “A simpler alternative would be to remove the gist token and directly use the hidden state of the last token to perform detection. However, intuitively, the hidden state of the last token is used for generation and may not encapsulate the information relevant to the harmfulness of the response.”

Comment: There are alternative approaches such as detection based on hidden states at the time of the first token generation, which one could argue is more efficient since we don’t have to wait for the token to be completely generated.

Page 5: “Like how the body detects and responds to pathogens, our system treats jailbreak activities as threats and uses a detector to identify them.”

Comment: Not sure how helpful this metaphor is since this is a standard detection and learn a classifier approach. The real question is: how do you avoid overfitting to the seen jailbreaking instructions?

Page 6: “... and Retokenization (Jain et al., 2023) and SmoothLLM (Robey et al., 2023) for LLM”

Comment: These baselines for LLMs are rather out-dated given the many more recent approaches, such as LLMScan and arXiv:2505.15753.

**Questions:**

Can you compare your method with recently proposed LLM jailbreak detection approaches?

---

### Official Review · Reviewer_9HAx · 2025-10-25

**Soundness:** 1
**Presentation:** 2
**Contribution:** 1
**Rating:** 2
**Confidence:** 5

**Summary:**

This paper proposes TTIM (Test-Time IMmunization), a defense framework that adaptively protects large language models (LLMs) and multimodal LLMs against jailbreak attacks during testing, rather than relying on pre-defined defense mechanisms.

**Strengths:**

1. Dynamic adaptive defense: TTIM continuously learns and adapts during testing, achieving 0% ASR after encountering only 10 jailbreak samples without requiring prior knowledge of attack methods.

2. Unified cross-modal framework: The method demonstrates consistent effectiveness across both LLMs and MLLMs against diverse attack types with minimal computational overhead.

**Weaknesses:**

**1. Limited Novelty: Core Idea Exists in Constitutional AI[1][2]**

**Fundamental approach not novel**: Detecting harmful outputs and performing self-alignment is Constitutional AI's established framework. The gist token for detection is primarily an engineering optimization rather than a conceptual breakthrough, and Constitutional AI's self-reflection may be more natural and better aligned with model distributions. The paper lacks clear differentiation from Constitutional AI's iterative self-improvement paradigm.

**2. Outdated Model Selection**

The experimental evaluation uses outdated models that limit the generalizability of findings. For LLMs, the paper only tests on LLaMA2-7B and LLaMA3-8B, missing current models like LLaMA3.2, Qwen2.5-7B, and Mistral-7B-v0.3. Similarly, for MLLMs, evaluation is limited to LLaVA-v1.6 variants and Qwen2VL-7B, while state-of-the-art models like Qwen2.5-VL, InternVL-3.5, and LLaVA-NeXT are absent. This raises serious questions about whether TTIM's effectiveness generalizes to modern models with improved safety alignments.

**3. Insufficient Attack and Defense Baselines**

The baseline coverage is severely imbalanced and incomplete, particularly for LLMs. While MLLM evaluation includes comprehensive attacks (Figstep, MM-SafetyBench) and defenses (Adashield, VLGuard), LLM evaluation only tests against I-FSJ and GCG, missing mainstream attack methods like AutoDAN[3], AmpleGCG[4], StrongREJECT[5] and WildJailbreak[6]. More critically, defense comparisons are limited to SmoothLLM and Retokenization, while key methods are absent: Constitutional AI (the most relevant comparison given the conceptual similarity noted in Weakness 1, Goal Prioritization[8] RAIN[7] and GradSafe[9] (relevant comparison that perform detection before the final output stage). This significant imbalance undermines the paper's claims of unified effectiveness across both LLMs and MLLMs.

[1] Constitutional ai: Harmlessness from ai feedback.

[2] Break the breakout: Reinventing LM defense against jailbreak attacks with self-refine.

[3] AutoDAN: Generating stealthy jailbreak prompts on aligned large language models.

[4] AmpleGCG: Learning a universal and transferable generative model of adversarial suffixes for jailbreaking both open and closed LLMs.

[5] A StrongREJECT for Empty Jailbreaks

[6] WildTeaming at Scale: From In-the-Wild Jailbreaks to (Adversarially) Safer Language Models

[7] RAIN: Your language models can align themselves without finetuning.

[8] Defending large language models against jailbreaking attacks through goal prioritization.

[9] GradSafe: Detecting unsafe prompts for LLMs via safety-critical gradient analysis

**Questions:**

1.  How does TTIM fundamentally differ from Constitutional AI's self-refinement mechanism?
2. Can the authors provide results on state-of-the-art models?
3. Why are mainstream LLM attack methods (AutoDAN, StrongREJECT, WildJailbreak) and critical defense baselines (especially Constitutional AI,goal prioritization,  RAIN) missing from the evaluation, and can comprehensive comparisons be added?

---

### Official Review · Reviewer_Vhk8 · 2025-10-30

**Soundness:** 2
**Presentation:** 2
**Contribution:** 3
**Rating:** 4
**Confidence:** 3

**Summary:**

The paper proposes TTIM, a test-time immunization framework that detects jailbreaks with a lightweight “gist” token and classifier, then fine-tunes a LoRA adapter online on detected jailbreak instructions paired with refusal responses. Detection and defense are parameter-decoupled to avoid loss of performance. Evaluated on MLLMs (LLaVA variants, Qwen2VL-7B) with Figstep and MM-SafetyBench, and on LLMs (LLaMA2-7B-chat, LLaMA3-8B-Instruct) with I-FSJ and GCG.  It achieves small ASR and ODR values while keeping inference overhead relatively low.

**Strengths:**

* The paper presents a simple and deployable mechanism in which a single learned “gist” token and lightweight classifier enable fast, low-cost jailbreak detection without requiring a separate guard model.
* The design supports online, adaptive defense using parameter-decoupled LoRA updates, allowing the model to improve its safety behavior during deployment while minimizing drift and preserving modularity.
* The approach achieves consistently low ODR relative to other baselines.
*  It includes a clear and favorable cost analysis demonstrating competitive overhead.

**Weaknesses:**

1. While it is true that many defense strategies rely on auxiliary LLMs or input augmentation, there are other detection approaches (e.g., classifier-based log-likelihood deviation detectors or anomaly scoring methods) [1-4] that should be acknowledged and compared against.
2. The LLM experiments are limited to the Llama family, including additional model families would strengthen the claims.
3. Although the proposed method is presented as one that “learns as it sees new samples,” the paper lacks numerical evidence showing robustness to more advanced or subtle attacks (for example, tool-invocation chains, hidden prompt injection, or meta-instruction pivot attacks) [5-7].
4. The rejection counting metric can be misleading because a refusal phrase may normally appear during a safe response. So, the reported ASR and ODR values may not reliably reflect actual performance. A more robust evaluation, such as using an auxilary LLM  for detecting harnmfullness should be used.
5. Additional results in Appendix B are not sufficiently explained. They lack detailed explanation, do not clearly benchmark against baselines, and it is hard to interpret them meaningfully.

***Minor remarks:***
1. Table captions could be more detailed to improve clarity and help readers better understand the reported results.
2. Highlighting the best performing values in each table would make comparisons more intuitive and visually more clear.
3. In line 373, the reference should point to Tables 1–2 rather than Tables 1–3.
4. Separating the analysis of LLMs and VLMs into distinct paragraphs would enhance readability.
5. The GCG results currently located in the appendix are important to the paper’s contributions and should be moved into the main text.

[1] Chen, G., Xia, Y., Jia, X., Li, Z., Torr, P., Gu, J. (2025) LLM Jailbreak Detection for (Almost) Free!

[2] Galinkin, E., Sablotny, M. (2024). Improved Large Language Model Jailbreak Detection via Pretrained Embeddings.

[3] Xie, Y., Fang, M., Pi, R., Gong, N. (2024). GradSafe: Detecting Jailbreak Prompts for LLMs via Safety-Critical Gradient Analysis.

[4] Hu, X., Chen, P.-Y., Ho, T.-Y. (2024). Gradient Cuff: Detecting Jailbreak Attacks on Large Language Models by Exploring Refusal Loss Landscapes.

[5] Andriushchenko, M., Croce, F., Flammarion, N. (2025). Jailbreaking Leading Safety-Aligned LLMs with Simple Adaptive Attacks.

[6] Chao, P., Robey, A., Dobriban, E., Hassani, H., Pappas, G.J., Wong, E. (2023). Jailbreaking Black Box Large Language Models in Twenty Queries.

[7] Mehrotra, A., Zampetakis, M., Kassianik, P., Nelson, B., Anderson, H., Singer, Y., Karbasi, A. (2023). Tree of Attacks: Jailbreaking Black-Box LLMs Automatically.

**Questions:**

1. My understanding is that the detector is trained only on benign and harmful sentences, rather than true jailbreak examples. Could you clarify whether jailbreak-style prompts appear in the training data at all?
2. Do the harmful goals used to generate adversarial attacks overlap with sentences present in the training set? For example, if “how to make a bomb?” appears during training, could that create unintended leakage? It would be helpful to evaluate on fully seperate harmful sets as a control.
3. Could you provide an ablation examining how the maximum memory size influences performance? The paper currently uses a limit of 40 examples; what motivated that specific choice?
4. How are the jailbreak attack samples collected or generated? Are they entirely self-generated or sourced from existing benchmarks or public datasets?
5. For each table, over how many evaluation samples are the ASR values computed? Please include those counts for transparency.
6. The simultaneous use of ASR, ODR, TPR, and FPR can be difficult to interpret. Could you more clearly differentiate TPR from (1 – ASR) and provide guidance on when each metric should be the primary reference?
7. In line 395, the statement that adaptation requires “fewer than 10 samples” is ambiguous. Could you include concrete numbers or adaptation-curves for each dataset and model?
8. The ODR of the vanilla model reported in Table 9 appears inconsistent with Table 2. Since ODR should not depend on the attack strategy applied, could you explain this discrepancy?

---

### Official Review · Reviewer_xvrh · 2025-10-31

**Soundness:** 3
**Presentation:** 3
**Contribution:** 2
**Rating:** 4
**Confidence:** 3

**Summary:**

This paper proposed a defense technique against jailbreaking attacks in  LLMs, called Test-Time Immunization (TTIM), which can adaptively defend against various jailbreak attacks through a self-evolving mechanism during inference.

**Strengths:**

1.	The paper introduces a defense technique against jailbreaking attacks against both LLMs and VLMs.
2.	This paper introduces a new defense technique that can continuously learn to defend against various jailbreak attacks by itself during inference.

**Weaknesses:**

1.	The paper includes investigations on a limited number of SOTA models, mostly relying on the open-sourced models. The lack of inclusion of commercial models, GPT-5, Claude, and Gemini 2.5, and reasoning models, o3 or R1 models, should be included in the evaluation.
2.	According to the definitions of TTIM provided in the paper and compared with the biological immune system, in the real world, how effective is the defense against emerging jailbreaking attacks in the first place?
3.	In lines 266-267, the immunity becomes stronger for future threats. How effective is it for the current or past threats?
4.	In table 1, Why the performance on LLaVa -13B model is not as strong as 7B? Appropriate explanation is required.
5.	The proposed defense is tested against a limited number of attacks. More recent attacks, e.g., MMJBench, Shuffle Inconsistency, IDEATOR, and BAP, these attacks should have been evaluated to better generalize the performance of the TTIM against jailbreak attacks.
6.	What’s the defense-time trade-off for the proposed defense?
7.	The paper should also discuss the failure cases of the TTIM against the attacks and the potential reasons behind those failure cases.
8.	Does the method fine-tune the target LLMs during test-time training phase?

**Questions:**

Please follow the Weaknesses.

---

### Official Review · Reviewer_Pmkn · 2025-11-01

**Soundness:** 3
**Presentation:** 3
**Contribution:** 3
**Rating:** 4
**Confidence:** 2

**Summary:**

This paper focuses on the jailbreak defense of LLMs and propose a self-evolving framework called TTIM. TTIM is a test-time defense method that does not require finetuning the target model and thus can apply to most LLMs. This paper also designs a low-cost, high-efficiency detector using gist tokens, avoiding reliance on auxiliary models. A parameter decoupling strategy is introduced to ensure the stability of both detection and defense training.

**Strengths:**

The method proposed in this paper does not rely on overly restrictive assumptions, enabling its application to a broad range of models.

**Weaknesses:**

The main weakness of this paper is that the models employed in the experiments and the comparative methods are outdated. Comparing the proposed method with the SOTA methods would enable a more rigorous and reliable evaluation of its effectiveness.

**Questions:**

See the weakness part.

---

### Note · Authors · 2025-12-09

I have read and agree with the venue's withdrawal policy on behalf of myself and my co-authors.